# PRIM2 Promotes Cell Cycle and Tumor Progression in p53-Mutant Lung Cancer

**DOI:** 10.3390/cancers14143370

**Published:** 2022-07-11

**Authors:** Taoyuan Wang, Tiansheng Tang, Youguo Jiang, Tao He, Luyu Qi, Hongkai Chang, Yaya Qiao, Mingming Sun, Changliang Shan, Xinyuan Zhu, Jianshi Liu, Jiyan Wang

**Affiliations:** 1Clinical School of Thoracic, Tianjin Medical University, Tianjin 300070, China; springwater1980@163.com (T.W.); tts20210819@163.com (T.T.); 2Department of Lung Cancer Surgery, Tianjin Medical University General Hospital, Tianjin 300052, China; jyouguo811@163.com; 3Departments of Pathology, Characteristic Medical Center of the Chinese People’s Armed Police Force, 220 Chenglin Road, Dongli District, Tianjin 300162, China; hetao_1981@163.com; 4Cardiothoracic Surgery Department, Characteristic Medical Center of the Chinese People’s Armed Police Force, 220 Chenglin Road, Dongli District, Tianjin 300162, China; qiluyu@wj120.cn; 5State Key Laboratory of Medicinal Chemical Biology, College of Pharmacy and Tianjin Key Laboratory of Molecular Drug Research, Nankai University, Tianjin 300350, China; 2120201229@mail.nankai.edu.cn (H.C.); 2120191147@mail.nankai.edu.cn (Y.Q.); 15522131992@163.com (M.S.); changliangshan@nankai.edu.cn (C.S.); 6Department of Cardiovascular Surgery, Tianjin Medical University General Hospital, Tianjin 300052, China; kuailezhuxinyuan@163.com; 7Department of Cardiovascular Surgery, Tianjin Chest Hospital, Tianjin 300222, China

**Keywords:** p53 mutation, cell cycle, mismatch repair, PRIM2, lung cancer

## Abstract

**Simple Summary:**

The mutation or inactivation of tumor suppressor genes is a key driving force during tumorigenesis, among which, p53 mutation is a common feature of human cancer. Therefore, exploring the potential role of p53 mutation in the occurrence and development of tumors is a powerful support for tumor diagnosis and treatment. In this study, we found that PRIM2 expression was abnormally elevated in p53-mutated lung cancer patients, and the elevated PRIM2 promoted DNA replication, enhanced mismatch repair, activated cell cycle, and promoted lung cancer progression. Here, we first report that the expression of PRIM2 is regulated by p53, and is identified as a biomarker of lung cancer malignancy and survival prognosis.

**Abstract:**

p53 is a common tumor suppressor, and its mutation drives tumorigenesis. What is more, p53 mutations have also been reported to be indicative of poor prognosis in lung cancer, but the detailed mechanism has not been elucidated. In this study, we found that DNA primase subunit 2 (PRIM2) had a high expression level and associated with poor prognosis in lung cancer. Furthermore, we found that PRIM2 expression was abnormally increased in lung cancer cells with p53 mutation or altered the p53/RB pathway based on database. We also verified that PRIM2 expression was elevated by mutation or deletion of p53 in lung cancer cell lines. Lastly, silence p53 increased the expression of RPIM2. Thus, these data suggest that PRIM2 is a cancer-promoting factor which is regulated by the p53/RB pathway. The p53 tumor-suppressor gene integrates numerous signals that control cell proliferation, cell cycle, and cell death; and the p53/RB pathway determines the cellular localization of transcription factor E2F, which regulates the expression of downstream targets. Next, we explored the role of PRIM2 in lung cancer and found that knockdown of PRIM2 induced cell cycle arrest, increased DNA damage, and increased cell senescence, leading to decreased lung cancer cell proliferation. Lastly, the positive correlation between PRIM2 and E2F/CDK also indicated that PRIM2 was involved in promoting cell cycle mediated by p53/RB pathway. These results confirmed that the expression of PRIM2 is regulated by the p53/RB pathway in lung cancer cells, promotes DNA replication and mismatch repair, and activates the cell cycle. Overall, we found that frequent p53 mutations increased PRIM2 expression, activated the cell cycle, and promoted lung cancer progression.

## 1. Introduction

Tumorigenesis is often as a result of uncontrolled cell growth and proliferation. In the runaway process, either abnormal activation of proto-oncogenes or inactivation of tumor suppressor genes occurs. The p53 gene was the first tumor suppressor gene to be identified, and the discovery that p53 has a role in human cancers has greatly facilitated the research of p53 functions in tumors [1]. In order to avoid accidents such as cell carcinogenesis, tumor growth, cell proliferation, and even death of normal cells, it is strictly controlled. Thus, p53 plays an important regulatory role in these processes [2]. p53 is more like a brake in the cell cycle. It will block the cell cycle or even induces cell apoptosis once the cells get out of control. Therefore, p53 is often mutated or inactivated in most tumors to ensure rapid cell proliferation. In fact, p53 is also “turned off” in normal cells and does not affect normal cellular processes. Only under certain stimuli (like DNA damage) will p53 be activated and involved in DNA damage repair, cell senescence, apoptosis, and other processes. In all, the mutation or inactivation of the tumor suppressor p53 is an important factor in tumorigenesis, which directly or indirectly regulates the abnormality of downstream pathways in tumors.

Lung cancer is the most common cancer and the most common cause of cancer death in the world [3]. The prognosis of lung cancer is poor, and the clinical outcomes are not satisfactory [4], so early detection will improve the prognosis of lung cancer [5,6]. Therefore, it is necessary to develop new biomarkers to support the early diagnosis and clinical treatment of lung cancer. In lung cancer, the most common alterations in tumor suppressor genes are mutations in *p53* [7]. At least half of lung cancer patients have *p53* mutations or deletions [8], suggesting that p53 dysfunction is a crucial factor in the development of lung cancer. More importantly, the mutation of *p53* is a negative prognostic factor for lung cancer [9], which further proves that p53 plays a key function in the progression of lung cancer. In addition, p53 is also involved in the radiotherapy and chemotherapy treatment of lung cancer [10]. Given that p53 and chemo-radiotherapy are involved in the DNA damage repair process, the drug resistance of lung cancer is also associated with p53. In addition to p53, the retinoblastoma (RB) gene is also a commonly inactivated gene in lung cancer [11,12]. It has been reported that RB is a potential therapeutic target for lung cancer [13]. Mutation or inactivation of tumor suppressor genes is a driver of tumor development, and these genes cannot be bypassed, whether to identify biomarkers for early diagnosis or as drug targets for therapy.

The cell cycle is mainly regulated by two pathways: p53 and RB signaling pathways [14]. Surprisingly, both pathways are inactivated at high frequency in lung cancer [7,8,11,12], suggesting that the inactivation of these two signaling pathways promotes the development of lung cancer. In the RB signaling pathway, extracellular growth stimuli activate receptor proteins, which in turn increase cyclin proteins expression. Additionally, the increased cyclin proteins bind to cyclin dependent kinase, which together phosphorylate the RB protein. These events promote E2F entry into the nucleus, which acts as a transcription factor to regulate downstream gene expression. In contrast, the p53 signaling pathway mediates the G1 checkpoint and exerts opposite regulatory roles. When events that are not suitable for cell division, such as DNA damage, occur in cells, p53 induces increased p21 expression, inhibits the activity of CDKs, and prevents cells from entering the division phase. These are sufficient to demonstrate the critical roles of RB and p53 pathways in the cell cycle. We have reason to suspect that the progression of lung cancer is inextricably linked with mutations in these two pathways. However, the specific roles of p53 and RB mutations in progress of lung cancer have not been fully explained.

Mitosis is an important process of cell division and is closely related to the progression of cancer. DNA replication is an early process in mitosis, guaranteeing that genetic information is passed on from generation to generation. Therefore, ensuring the accuracy and efficiency of DNA replication is very necessary for cell division. During DNA replication, the double helix DNA is unwound, and the complementary strands are separated by helicases, forming the DNA replication fork. Since the synthesis direction of DNA polymerase is 5′ to 3′ and the two strands of DNA are anti-parallel, the two chains are not replicated in the same way. Replication of lagging strands is discontinuous and requires the formation of Okazaki fragments and the involvement of PRIM2. So far, little research has been done on the mechanism of PRIM2 in lung cancer, and it is mainly found that PRIM2 affects the survival of lung cancer patients [15] and participates in ferroptosis [16].

In this study, we found that high expression of PRIM2 was associated with poor prognosis in lung cancer patients. Gene set enrichment analysis (GSEA) found that PRIM2 may promote the progression of lung cancer by mediating cell cycle and DNA damage repair. Further analysis found that the expression abundance of PRIM2 was related to the status of p53. Specifically, PRIM2 expression was upregulated when p53 was mutated. In conclusion, we found that PRIM2 has an important regulatory role in the progression of lung cancer, which will provide theoretical support for the clinical diagnosis and treatment of lung cancer, including new biomarkers for the early diagnosis of lung cancer and potential therapeutic targets.

## 2. Materials and Methods

### 2.1. The Analysis for Differential Gene Expression and Mutation

The mRNA expression and protein expression data were obtained from The Cancer Genome Atlas (TCGA: http://ualcan.path.uab.edu/analysis (accessed on 20 April 2022)) [17] and Clinical Proteomic Tumor Analysis Consortium (CPTAC: http://ualcan.path.uab.edu/analysis-prot (accessed on 20 April 2022)) [18]. The somatic mutation data were obtained from TCGA cBioportal platform (https://www.cbioportal.org/ (accessed on 20 April 2022)) [19,20]. The mutations were identified and analyzed in the cBioportal platform.

### 2.2. Gene Expression Omnibus

The RNA sequencing data were collected from the Gene Expression Omnibus (GEO, https://www.ncbi.nlm.nih.gov/geo/ accessed on 5 May 2022) datasets: GSE19804 [21], GSE103512 [22], GSE47436 [23], GSE184414, and GSE87879 [24].

### 2.3. Survival Prognostic Analysis

The prognostic analysis of patients was grouped by different gene expression based on the Kaplan–Meier Plotter (http://kmplot.com/analysis/index.php?p=background accessed on 21 April 2022) [25].

### 2.4. Gene Set Enrichment Analysis

Gene set enrichment analysis (GSEA) was employed using GSEA 4.0.3 (Broad Institute, Cambridge, MA, USA) (http://software.broadinstitute.org/gsea/index.jsp (accessed on 21 April 2022)). We divided lung cancer patients into two groups according to the expression of PRIM2, including a high-expression group and low-expression group.

### 2.5. Correlation Analysis

Expression of the indicated genes was obtained from the TCGA database, and correlation analysis was performed using GraphPad Prism 8 (GraphPad Software Inc., San Diego, CA, USA). A Pearson coefficient (r value) greater than zero represents a positive correlation, and less than zero represents a negative correlation. The larger the absolute value of r, the stronger the correlation.

### 2.6. Immunohistochemistry

The lung cancer tissues were collected from Characteristic Medical Center of The Chinese People’s Armed Police Force (Tianjin, China). The tumor tissues included 27 lung cancer tissues and 27 adjacent tumor tissues. We performed immunohistochemistry (IHC) assays according the protocol in a previously published article [26]. Briefly, the paraffin-embedded sections were dewaxed and rehydrated. Antigen retrieval was performed by boiling samples in citrate buffer. Endogenous peroxidase activity was inhibited by using hydrogen peroxidase. Sections were blocked in 3% normal goat serum in PBS and incubated in primary antibody. Sections were rinsed in PBS and developed using DAB. Sections were counterstained with hematoxylin. Quantification of staining was performed using Image J software.

### 2.7. Cell Culture

The human lung cancer cell lines (H460, A549, H1299, H2122, and H1155) were grown in RPMI1640 containing 10% (*v*/*v*) fetal bovine serum (FBS, ExCell Bio, Taichang, China) and 1% (*v*/*v*) penicillin/streptomycin. The human embryonic kidney cell line HEK293T was maintained in DMEM supplemented with 10% (*v*/*v*) fetal bovine serum (FBS, ExCell Bio, China) and 1% (*v*/*v*) penicillin/streptomycin. All cells were cultured at 37 °C in an incubator supplied with 5% CO_2_. To generate PRIM2 knockdown cell lines, lentivirus was generated by co-transfecting HEK293T cells with pLKO.1-shPRIM2, envelop plasmid pMD2.G, and packaging plasmid psPAX2 using a polyethylenimine (PEI) transfection reagent (Polysciences, Warrington, PA, USA), following the manufacturer’s protocol. H1299 cells were then infected with filtered lentiviral supernatant and selected with 5 μg/mL of puromycin.

### 2.8. Real-Time PCR

Total RNA was isolated with the TRIzol reagent, and then was subjected to reverse transcription using the PrimeScript RT reagent Kit with gDNA Eraser (RR047A, Takara). RT-qPCR reactions were performed with the TB Green™ Premix Ex Taq™ II (RR820A, Takara). Gene expression was calculated using the comparative 2^−ΔΔCT^ method with actin for normalization. The primers were as Table 1:

### 2.9. Western Blot

The cell pellets were lysed in the RIPA lysis buffer on ice for 20 min. The cell lysate was centrifuged, and an equal amount of cell lysate from each sample was loaded onto the gels for SDS-PAGE. The protein was transferred onto PVDF membranes (Millipore, Burlington, MA, USA), blocked with 5% nonfat milk, and incubated with primary antibodies against CDK4 (11026-1-AP, Proteintech) and β-actin (66009-1-Ig, Proteintech). HRP conjugated goat anti-mouse or anti-rabbit IgG was used as secondary antibody. The signal was detected by Immobilon Western HRP Kit (Millipore, Burlington, MA, USA).

### 2.10. Cell Growth Assays and Cell Senescence Assays

For cell growth assay, PRIM2 knockdown and vector control lung cancer cells were seeded in 24-well plates. On the second day of seeding, the counting was started and lasted four days. For cell senescence assay, a SA-β-gal staining kit was used to stain senescent cells. According to the manufacturer’s instructions, cells were fixed for 15 min, stained for 24 h and imaged. Senescent cells, identified as blue-stained cells, were captured with light microscopy.

### 2.11. Statistics

Data analysis and charting were performed using GraphPad Prism 8 (GraphPad Software Inc., San Diego, CA, USA). The data are shown as mean ± standard deviation (S.D.). Statistical significance was analyzed using a two-tailed Student’s *t*-test, and *p* < 0.05 was considered statistically different.

## 3. Results

### 3.1. PRIM2 Is Upregulated in Lung Cancer with Poor Prognosis

To explore the roles of PRIM2 in lung cancer, we first analyzed the expression of PRIM2 in lung cancer and found that PRIM2 was significantly upregulated in lung adenocarcinoma (LUAD) at the mRNA and protein levels (Figure 1A,B). The same results were observed in lung cancer patients from the GEO database (Figure 1C). To validate the clinical relevance of our findings based on public databases, we examined the expression of PRIM2 in tumor tissue, which was collected from the Characteristic Medical Center of The Chinese People’s Armed Police Force. We found that PRIM2 was significantly elevated in tumor tissue relative to normal tissues by IHC (Figure 1D,E). In addition, we also found that the expression of PRIM2 gradually increased as the stage of LUAD increased (Figure 1F), which suggests a potential connection between PRIM2 and LUAD progression. To further explore role of PRIM2, we knocked down PRIM2 in lung cancer H1299 cells and found that knockdown of PRIM2 inhibited the growth of lung cancer cells (Figure 1G,H), which shows PRIM2 promotes lung cancer progression. To probe whether high expression of PRIM2 possesses diagnostic significance for lung cancer patients, the receiver operator characteristic (ROC) curves were created to analyze the diagnostic value of PRIM2 expression from TCGA-LUNG datasets. ROC curve analysis revealed that PRIM2 could statistically distinguish lung cancer patients from normal individuals, producing an area under the curve (AUC) of 0.904 (95% CI: 0.880–0.927; *p* < 0.0001) (Figure 1I). These results strongly suggest that PRIM2 might have diagnostic value for patients with lung cancer. To this end, we wonder whether the expression level of PRIM2 was associated with the prognosis of lung cancer. The results show that the high expression of PRIM2 was significantly associated with poor prognosis in patients (Figure 1J). Together, these findings demonstrate that PRIM2 expression was elevated in lung cancer and was correlated with poor patient prognosis, suggesting that PRIM2 has the potential to be a biomarker for lung cancer.

### 3.2. PRIM2 Promotes Cell Cycle, DNA Replication, and Mismatch Repair through PCNA in Lung Cancer

To explore the molecular biological function of PRIM2 upregulation in lung cancer, we firstly divided the lung cancer patients sample from the TCGA database into two groups (high PRIM2 expression group and low PRIM2 expression group) according to the expression of PRIM2. Using gene set enrichment analysis (GSEA), we found that highly expressed PRIM2 promotes the cell cycle, DNA replication, and mismatch repair (Figure 2A). Additionally, CDK4 expression was significantly reduced in lung cancer cells with PRIM2 knockdown (Figure 2B), indicating the cell cycle is suppressed. This result is consistent with the inherent primerase activity of PRIM2. When the cell cycle is active, large amounts of DNA are synthesized and distributed to daughter cells. PRIM2 acts as a powerful helper for the replication lagging strand, facilitating the DNA replication process. Unexpectedly, PRIM2 also promotes mismatch repair, a foremost repair mechanism during the cell cycle. As mismatch repair needs to distinguish parent and daughter strands, it avoids excision of the correct nucleotide on the parent strand.

To identify downstream targets regulated by PRIM2, we performed overlap analysis and found that proliferating cell nuclear antigen (PCNA) may be a potential target (Figure 2C). Indeed, we found that PCNA expression was also significantly increased in lung cancer patients’ tumors from TCGA datasets and the GEO database (Figure 2D,E). In addition, in our study, we found a positive correlation between the expression levels of PCNA and PRIM2, suggesting that the expression of PCNA may be regulated by PRIM2 (Figure 2F). To this end, we found that PCNA expression was significantly downregulated in PRIM2 knockdown cells (Figure 2G), indicating that PCNA expression is regulated by PRIM2. Both cell cycle arrest and DNA damage induce cell senescence. As expected, knockdown of PRIM2 caused severe cellular senescence in lung cancer cells (Figure 2H). In all, our results showed that highly expressed PRIM2 exerts primerase activity to promote DNA replication, mismatch repair, and the cell cycle through regulating PCNA.

### 3.3. High PRIM2 Expression Is Dependent on Mutation of p53

The mutations in tumor suppressor genes are a major driver of tumorigenesis. Thus, we wanted to explore the mechanism underlying the high expression PRIM2 in lung cancer from the tumor suppressor angle. We divided lung cancer patients into p53 mutant and non-mutant groups, and found that the expression levels of PRIM2 and PCNA were significantly increased in the p53 mutant group (Figure 3A), suggesting that PRIM2 and PCNA expression may be partially regulated by p53. We have previously found that the function of PRIM2 is to activate the cell cycle, which is regulated by the p53 signaling pathway and the RB signaling pathway. When p53 and RB pathways were altered in lung cancer cells, the protein levels of PRIM2 and PCNA were elevated (Figure 3B). Therefore, we speculated that PRIM2 expression is regulated by the p53 and RB pathways.

To this end, we analyzed the mutation frequencies of TP53, RB1, PRIM2, and PCNA in LUAD. The results showed that TP53 had the highest mutation frequency, as high as 46%. RB1 also had a 7% mutation frequency, while PRIM2 and PCNA had lower mutation frequencies (Figure 3C). We also noticed a correlation between tumor mutational burden (TMB) and TP53 mutations, suggesting that TP53 mutations play an important role in lung cancer. To further prove the relationship between p53 mutation and PRIM2 expression, we detected the expression level of PRIM2 in different lung cancer cells. Compared with wild type, the expression of PRIM2 was significantly increased in p53 mutant or deficient cells (Figure 3D). Likewise, the expression of PRIM2 and PCNA was upegulated when p53 was knocked down in airway cells (Figure 3E). These results further confirmed that PRIM2 is regulated by p53.

### 3.4. The p53/RB Pathway–PRIM2–PCNA Axis Promotes Cell Growth by Regulating Cell Cycle

The previous results allowed us to determine that the expression of PRIM2 is regulated by p53 and is involved in the regulation of the cell cycle. The following analysis found that E2F and CDK family members had significant positive correlations with PRIM2 (Figure 4A,B). Furthermore, we found potential binding elements for transcription factors E2F1 and E2F6 on the PRIM2 promoter, suggesting that PRIM2 is under the transcriptional regulation of E2F (Figure 4C). In addition, treatment with cell-cycle-inhibiting small molecules (PD-0332991 and Berberine) induced downregulation of PRIM2 and PCNA expression, further suggesting the important role of PRIM2 in regulating cell cycle (Figure 4D,E). Collectively, PRIM2 and PCNA mediate the regulation of the cell cycle by p53/RB (Figure 4F).

## 4. Discussion

An uncontrolled cell cycle is an important feature that distinguishes tumor cells from normal cells. The proliferation of a cell is a process that involves DNA replication and then dividing everything in two to form two daughter cells; this process known as the cell cycle. Tumor development is driven by the inactivation of tumor suppressor genes that directly lead to abnormal activation of the cell cycle. Since rapid growth and proliferation are the most distinctive features of tumor cells, the cell cycle is activated through different pathways. We have previously showed that the abnormalities in tyrosine metabolizing enzymes also activate the cell cycle and promote tumor cells’ proliferation [27,28]. Therefore, whether identifying tumor markers or exploring new therapeutic targets, starting from the cell cycle is a promising research strategy.

PRIM2 is a key protein that plays an important role in DNA replication, but there are not many related studies. DNA replication is the most important step in the cell cycle, and it is critical to ensure daughter cells can maintain the continuity and stability of their genomes. It is well known that DNA is a double-stranded molecule. The double-helix is first opened, and then the replication fork is formed during DNA replication. However, DNA polymerase can only synthesize from 5′ to 3′; in this way, only the leading strand can be synthesized. For the lagging strand, Okazaki fragments need to be formed first, and then DNA ligase joins them into a complete strand. PRIM2 is responsible for the synthesis of Okazaki fragments, and determines the smooth progression of the cell cycle. To explore the role of PRIM2, we analyzed PRIM2 expression in lung cancer from the TCGA database. The results showed that the expression of PRIM2 is abnormally increased in lung cancer. In addition, PRIM2 expression increased with stage and was closely related to the poor prognosis of patients, which indicates that PRIM2 plays an important regulatory role in the occurrence and development of lung cancer. Although it has been found that PRIM2 antagonizes ferroptosis to play a cancer-promoting role in lung cancer [16], it has not been demonstrated that this role is dependent on the enzymatic activity of PRIM2. In the pathway enrichment results, we found that PRIM2 promotes DNA replication and mismatch repair, and activates the cell cycle. In the DNA replication process, PCNA mediates the function of PRIM2. The function of PCNA in lung cancer was found to be regulated by miRNA [29,30], which promotes the growth and proliferation of tumor cells, and PCNA is a potential therapeutic target. These results indicate that PRIM2, after expression upregulation in lung cancer, not only exerts its own primerase activity to promote DNA replication, but also induces an increase in PCNA expression, promotes mismatch repair, and promotes DNA replication. This is the first time that PRIM2 has been shown to regulate the cell cycle and play a cancer-promoting role in lung cancer, which will lay the foundation for the subsequent exploration of PRIM2.

The discovery of increased PRIM2 expression in patients with altered p53 and RB signaling pathways in lung cancer drew our attention. p53 and RB are well-known tumor suppressors, and they are often mutated and inactive in lung cancer [7,8,11,12]. Therefore, we speculate that the expression of PRIM2 may be regulated by the p53 and RB pathways. Both p53 and RB regulate the cell cycle. They are like the accelerator and the brakes of the cell cycle, regulating the cell cycle in both directions. The cell cycle’s co-regulation by p53 and RB is finally aggregated in the transcription factor E2F to regulate the expression of downstream genes. The expression levels of PRIM2 and E2F had a significant positive correlation, suggesting the regulation of E2F by PRIM2. While testing our hypothesis using immunohistochemical detection in lung cancer patients, PRIM2 was found to be highly expressed, and the high expression of PRIM2 in lung cancer cells was found to be related to the mutation of p53. Both PRIM2 and PCNA expression were altered when p53 was knocked down or CDK activity was inhibited. These results indicate that the expression abundance of PRIM2, which promotes cancer, is regulated by the p53/RB pathway. When we knocked down PRIM2 in lung cancer cells, we found that cell growth was inhibited, the cell cycle was arrested, and cellular senescence was promoted. These results demonstrate that PRIM2 is regulated by p53 while regulating downstream PCNA expression to promote DNA replication and cell cycle.

## 5. Conclusions

In conclusion, we identified PRIM2 as a potential biomarker and prognostic factor for lung cancer. Up to 46% of p53 mutations occur in lung cancer patients, which also means that PRIM2 expression is elevated in at least half of lung cancer patients. In this way, we can use PRIM2 as a therapeutic target to develop new anticancer drugs or therapeutic strategies in p53-mutated lung cancer patients. Mutation of p53 is a common feature of tumorigenesis, so it is of broad significance to study the mechanism of p53 mutation-induced tumorigenesis and development. Although we found PRIM2 in lung cancer to mediate a tumor-promoting effect after p53 mutation, this could extend to other tumors. Last, we must acknowledge that the cancer-promoting role of PRIM2 in lung cancer needs further experimental verification based on strict study.

## Figures and Tables

**Figure 1 cancers-14-03370-f001:**
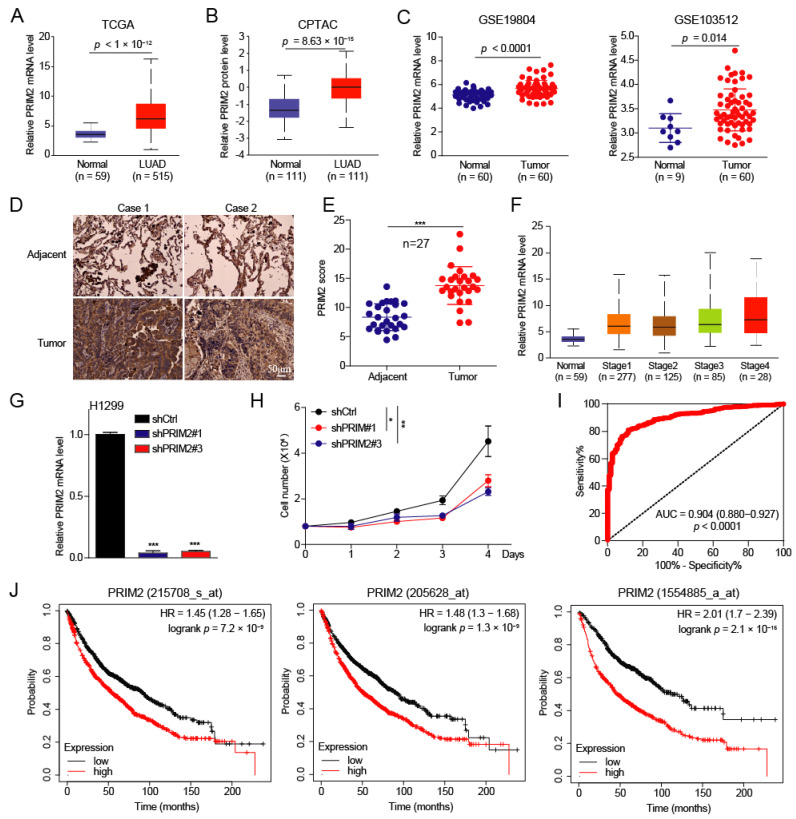
PRIM2 is upregulated and associated with prognosis in lung cancer. (**A**–**C**) The PRIM2 mRNA and protein expression levels were analyzed in tumor and normal tissues. (**D**,**E**) The expression of PRIM2 was examined in lung tumor and adjacent tissues of lung cancer patients by immunohistochemical (IHC) analysis. (**F**) The PRIM2 mRNA levels were analyzed for different stages of tumor and normal tissues. (**G**) The PRIM2 mRNA expression levels in H1299 cells with PRIM2 knockdown. (**H**) The cell growth in H1299 cells with PRIM2 knockdown. (**I**) ROC curve analysis indicated that PRIM2 could efficiently distinguish a lung cancer patient from a normal individual. (**J**) Kaplan–Meier analysis of overall survival was performed on the basis of PRIM2 expression. * *p* < 0.05; ** *p* < 0.01; *** *p* < 0.001.

**Figure 2 cancers-14-03370-f002:**
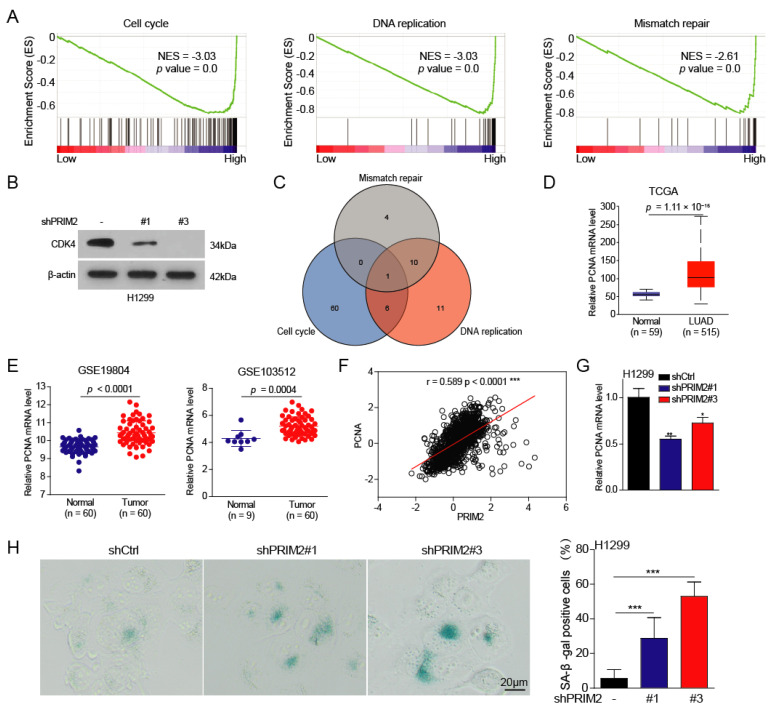
PRIM2 activates the cell cycle and promotes DNA repair in lung cancer. (**A**) GSEA analysis of the PRIM2 signature in patients with lung cancer. (**B**) The expression of CDK4 in H1299 cells with PRIM2 knockdown. (**C**) The overlapping analysis for target genes of the cell cycle, DNA replication, and mismatch repair in lung cancer patients. (**D**,**E**) The PCNA mRNA expression levels were analyzed for tumor and normal tissues. (**F**) The expression correlation between PRIM2 and PCNA. (**G**) The PCNA mRNA expression levels were examined in H1299 cells with PRIM2 knockdown. (**H**) Cell senescence assay was performed on H1299 cells with PRIM2 knockdown. * *p* < 0.05; ** *p* < 0.01; *** *p* < 0.001. The uncropped WB figures are in Appendix A.

**Figure 3 cancers-14-03370-f003:**
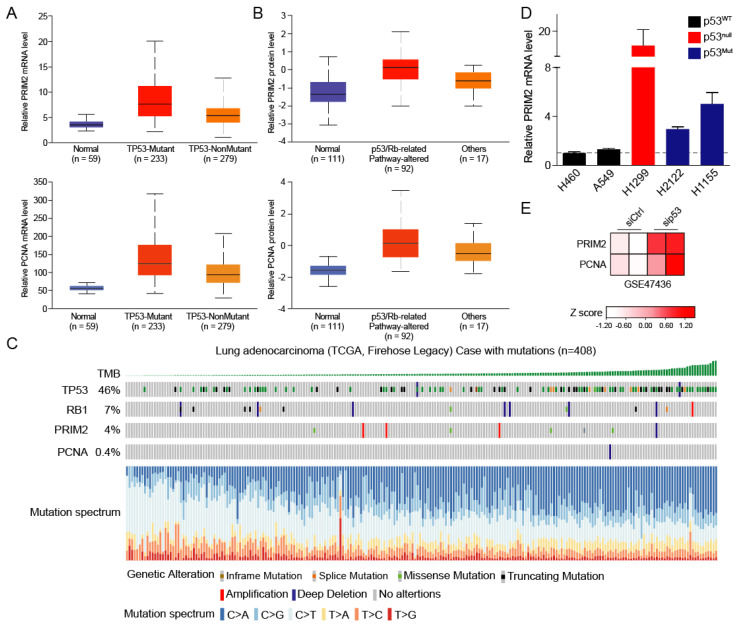
PRIM2 expression is associated with alteration of p53. (**A**,**B**) The PRIM2 mRNA and protein expression levels were analyzed among normal tissues, p53-mutant tissues, and p53-nonmutant tissues in patients with LUAD. (**C**) The types and frequencies of mutations of TP53, RB1, PRIM2, and PCNA were analyzed in LUAD patients. (**D**) The expression of PRIM2 was analyzed in different lung cancer cells. (**E**) The PRIM2 and PCNA mRNA expression levels were analyzed in airway cells with sip53.

**Figure 4 cancers-14-03370-f004:**
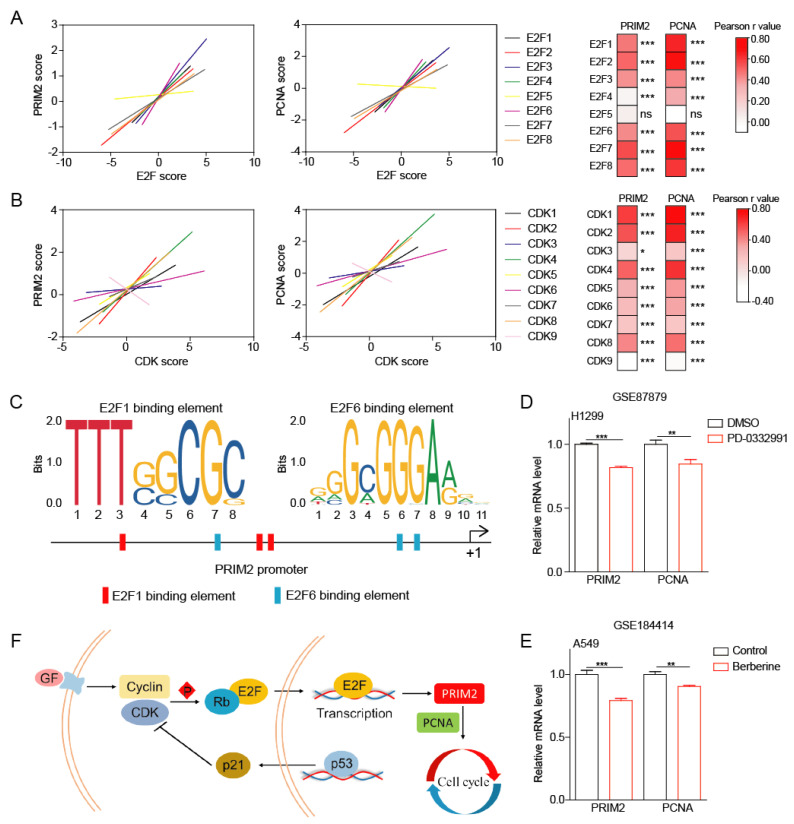
The p53/RB pathway–PRIM2–PCNA axis promotes cell growth. (**A**,**B**) The expression correlation between PRIM2 and E2F/CDK family members. (**C**) The predictive analysis of transcription factor E2F-binding elements on the PRIM2 promoter. (**D**) The PRIM2 and PCNA mRNA expression levels in H1299 cells with the treatment of PD-0332991 from the GEO database. (**E**) The PRIM2 and PCNA mRNA expression levels in A549 cells with the treatment of Berberine from the GEO database. (**F**) Schematic representation of PRIM2 regulation of the cell cycle. * *p* < 0.05; ** *p* < 0.01; *** *p* < 0.001.

**Table 1 cancers-14-03370-t001:** The sequence of primers for Real-time PCR.

Gene	Sequence (5′→3′)
PRIM2	AATGCTTCCTACCCTCATTGC
	AGCTCACTCTCCAACTTACTCTG
PCNA	CCTGCTGGGATATTAGCTCCA
	CAGCGGTAGGTGTCGAAGC
ACTIN	GGAAATCGTGCGTGACAT
	TGCCAATGGTGATGACCT

## Data Availability

All data will be made available upon reasonable request by emailing the corresponding author.

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
