# Peer review of "PRIM2 Promotes Cell Cycle and Tumor Progression in p53-Mutant Lung Cancer"

_cancers, 2022, doi:10.3390/cancers14143370_

Round 1

Reviewer 1 Report

In this manuscript, the authors characterized that role of PRIM2 in Lung cancer. They found that PRIM2 expression was increased in p53-mutated lung cancer patients, and elevated PRIM2 promoted cell cycle and DNA repair. In addition, the PRIM2 expression is dependent on P53 status. 

Overall, this finding is novel and interesting. However, only one IHC was performed for the whole manuscript. The claims are not fully supported by the experiments but by bioinformatics analysis. The authors are need to validate their finding by experiments. My enthusiasm is limited because of the following reasons:

Major Comments:

1.  The authors are required to stain PRIM2 in normal and tumor samples. They need to do qRT-PCR if there is no antibody available. 

2. The authors need to knockdown of PRIM2 in cells, and then perform cell cycle analysis by FACS and check DNA repair pathways by western blot. 

3. They need to perform western blot/qRT-PCR to measure PRIM2 in several p53 WT and p53 MU cell lines. 

4. Please describe more about PRIM2 in the introduction section. 

4. Moderate editing of English language is required. 

5. In Figure legend 1, PRIM2 is down-regulated?

Author Response

Comments and Suggestions for Authors 1:

In this manuscript, the authors characterized that role of PRIM2 in Lung cancer. They found that PRIM2 expression was increased in p53-mutated lung cancer patients, and elevated PRIM2 promoted cell cycle and DNA repair. In addition, the PRIM2 expression is dependent on P53 status.

Overall, this finding is novel and interesting. However, only one IHC was performed for the whole manuscript. The claims are not fully supported by the experiments but by bioinformatics analysis. The authors are need to validate their finding by experiments. My enthusiasm is limited because of the following reasons:

Major Comments:

  1. The authors are required to stain PRIM2 in normal and tumor samples. They need to do qRT-PCR if there is no antibody available.

Response: We appreciate the reviewer for pointing out this important issue. As suggested by the reviewers, we examined the expression levels of PRIM2 in tumor and adjacent tissues by IHC and found that PRIM2 expression was significantly elevated in tumor tissues (as shown below).

  1. The authors need to knockdown of PRIM2 in cells, and then perform cell cycle analysis by FACS and check DNA repair pathways by western blot.

Response: We appreciate the reviewer for pointing out this important issue. As suggested by the reviewers, we established the PRIM2 knockdown lung cancer cell line (H1299) (Figure A), and found the expression of PCNA is reduced in PRIM2 knockdown cells (Figure B). In addition, we also examined the expression of the cell cycle-related proteins CDK4 and DNA damage marker protein γH2AX and found that CDK4 was down-regulated, indicating that knockdown of PRIM2 resulted in cell cycle arrest (Figure C). However, we did not detect the expression of γH2AX, probably because DNA mismatches do not produce DNA double-strand breaks. Thus, we detected cell growth and cellular senescence in PRIM2 knockdown cells and found that cell growth inhibition and cellular senescence were induced after PRIM2 knockdown (Figure D and E). These results suggest that knockdown of PRIM2 leads to cell cycle arrest and DNA damage.

  1. They need to perform western blot/qRT-PCR to measure PRIM2 in several p53 WT and p53 MU cell lines.

Response: We appreciate the reviewer for pointing out this important issue. As suggested by the reviewers, we measure PRIM2 expression level by qRT-PCR in p53-WT and p53-Mut cell lines. As shown in the figure below, PRIM2 is more expressed in p53-mutated cell lines than in wild-type. And the expression of PRIM2 was also increased in deletion mutant H1299 cells, compared with wild-type cells.

  1. Please describe more about PRIM2 in the introduction section.

Response: We appreciate the reviewer for pointing out this important issue. We are sorry for that we did not make it clear in present manuscript. As suggested by the reviewers, we have added more information about PRIM2 in the introduction section.

  1. Moderate editing of English language is required.

Response: We appreciate the reviewer for pointing out this important issue. As suggested by the reviewers, we have edit English language.

  1. In Figure legend 1, PRIM2 is down-regulated?

Response: We appreciate the reviewer for pointing out this important issue. We are sorry for that we did make it wrong in present manuscript. We have revised it in the revised manuscript.

Reviewer 2 Report

11)      Line 57-59: ‘The growth, proliferation and even death of normal cells are strictly controlled, and p53 plays an important regulatory role in 58 these processes.’ It would be worth rewording this sentence, perhaps in this way: The growth, proliferation and even death of normal cells are strictly controlled, and by p53. plays an important regulatory role in these processes

22)      Line 59-60: ‘The p53 is more like a brake in the cell cycle, blocking or even killing abnormal cells once the cells get out of control.’ P53 does not kill cells, but is involved in stopping the cell cycle when DNA damage occurs, and if it can't be repaired, it is involved in directing the cell to the cell death pathway. This sentence should be rewritten.

33)      Line 72-73: ‘…p53 mutations or deletions.’ If we are already talking about a deletion, we implicitly know that it is a gene, so the notation p53 should be in italics.

44)      Line 97-99: ‘Therefore, dysregulation of p53 and RB pathways is likely to induce cell cycle abnormalities in lung cancer cells, and the cell cycle is a promising target for clinical cancer treatment, but its detailed mechanism has not been elucidated.’ Which mechanism? One can guess that the authors' point is that despite intensive research into the TP53 and RB1 pathways and really a lot of findings, there is still no effective molecularly targeted therapy developed for these targets. Generally, the authors include quite a bit of superficial information in the introduction, just the type that you can guess what the authors are talking about, but the verbal record of the information is somewhat superficial.

55)      Line 100-105: ‘In this study, we found that the expression of PRIM2 was significantly increased in lung cancer, and the prognosis of lung cancer patients with high expression was poor, which indicated that the high expression of PRIM2 increased the malignant degree of lung cancer. Further analysis found that the highly expressed PRIM2 exerts its primerase activity to promote the DNA replication of tumor cells and activate the cell cycle. In addition, PRIM2 also promotes mismatch repair that occurs during DNA replication to ensure smooth replication. Among different classifications of lung cancer, we found that the expression of PRIM2 was significantly up-regulated in lung cancers with altered p53 and RB signaling pathways, suggesting that PRIM2 may be regulated by p53 and RB pathways’ It looks like a description of the results, relatively part of the discussion, and there is this paragraph in the introduction. Is this the way it should be?

66)      Line 109-111: if it is liteterure report, citation is missing.

77)      Line 113-119: again, wording and summaries specific to the results or discussion.

88)      Line 152: ‘2.6. Immunohistochemistry’: How many patients were tested? Where did the study material come from? Was the study prospective or retrospective? The expression of which proteins was determined by this method? Was there approval for the study from the appropriate bioethics committee?

99)      Line 167-176: This paragraph is an introduction and not a description of the results.

110)   Line 183-184: ‘…which may suggest that the abnormal expression of PRIM2 protein is caused by the increased transcription level.’ Again, a phrase that fits the discussion not the results. Further part of the results also contain wording charateristic to the discussion, not the results.

111)   Line 190: KIRC – no expansions of the abbreviation, the same line 218 PCNA . Throughout the article: if authors use abbreviations, the full name of the abbreviation should be given the first time it is used.

112)   Line223-225: ‘PCNA is a cofactor for the DNA polymerase delta localized in the nucleus, increases the processivity of the leading strand during DNA replication, and is also involved in DNA repair. Research has found that PCNA is regulated by miRNAs to promote the growth and proliferation of lung cancer [23, 24].’ Once again, sentences specific to the introduction, not the results.  The same line 260-262.

113)   The authors identified p53 mutant and non-mutant groups in their analyses. What were the selection criteria? What variants were considered? Were these mutations of pathogenic status, or was selection based on the functional type of mutation? There is no such explanation in the text.

114)   Line 276-282: The entire paragraph is described as in the discussion. Only here does the reader learn that TP53 expression was determined using IHC.

115)   And returning to point 13 of this review, it is only in the course of reading and comparing the graphs that the reader can infer for himself how the selection of patients was done in terms of TP53 mutations. The authors completely overlooked this in the methodology. 

116)   Line 309: ‘Cell growth and proliferation are required for normal cells and tumor cells.’ It is not entirely clear what the authors intended to bring to the discussion with this sentence. Perhaps the authors would consider starting the discussion differently? How about eliminating The following sentences are more precise.

117)   The authors frequently mention the role of TP53 and RB1 pathways in the article, but did not perform any analysis of RB1. This is somewhat lacking. At the same time, it is understandable that analyses have their limitations in quantity. Nevertheless, RB1 was dragged into the topic of the article, and it would have been nice if there had been an analysis at least at the RB1 mRNA level, if it is often mentioned in the article.

118)   The figures are of very good quality, are legible and easily interpretable. Nice.

Author Response

Comments and Suggestions for Authors 2:

1) Line 57-59: ‘The growth, proliferation and even death of normal cells are strictly controlled, and p53 plays an important regulatory role in 58 these processes.’ It would be worth rewording this sentence, perhaps in this way: The growth, proliferation and even death of normal cells are strictly controlled, and by p53 plays an important regulatory role in these processes.

Response: We appreciate the reviewer for pointing out this important issue. We are sorry for that we did not make it clear in present manuscript. As suggested by the reviewers, we have revised it in the revised manuscript.

2) Line 59-60: ‘The p53 is more like a brake in the cell cycle, blocking or even killing abnormal cells once the cells get out of control.’ P53 does not kill cells, but is involved in stopping the cell cycle when DNA damage occurs, and if it can't be repaired, it is involved in directing the cell to the cell death pathway. This sentence should be rewritten.

Response: We appreciate the reviewer for pointing out this important issue. We are sorry for that we did not make it clear in present manuscript. As suggested by the reviewers, we have revised it in the revised manuscript.

3) Line 72-73: ‘…p53 mutations or deletions.’ If we are already talking about a deletion, we implicitly know that it is a gene, so the notation p53 should be in italics.

Response: We appreciate the reviewer for pointing out this important issue. We are sorry for that we did not make it clear in present manuscript. As suggested by the reviewers, we have revised it in the revised manuscript.

4) Line 97-99: ‘Therefore, dysregulation of p53 and RB pathways is likely to induce cell cycle abnormalities in lung cancer cells, and the cell cycle is a promising target for clinical cancer treatment, but its detailed mechanism has not been elucidated.’ Which mechanism? One can guess that the authors' point is that despite intensive research into the TP53 and RB1 pathways and really a lot of findings, there is still no effective molecularly targeted therapy developed for these targets. Generally, the authors include quite a bit of superficial information in the introduction, just the type that you can guess what the authors are talking about, but the verbal record of the information is somewhat superficial.

Response: We appreciate the reviewer for pointing out this important issue. We are sorry for that we did not make it clear in present manuscript. As the reviewer stated, we would like to express that there are many studies finding that the p53 and RB pathways play an important role in tumors, but the factors that are regulated by them have not been fully explained. We have revised it in the revised manuscript to make it more straightforward to understand.

5) Line 100-105: ‘In this study, we found that the expression of PRIM2 was significantly increased in lung cancer, and the prognosis of lung cancer patients with high expression was poor, which indicated that the high expression of PRIM2 increased the malignant degree of lung cancer. Further analysis found that the highly expressed PRIM2 exerts its primerase activity to promote the DNA replication of tumor cells and activate the cell cycle. In addition, PRIM2 also promotes mismatch repair that occurs during DNA replication to ensure smooth replication. Among different classifications of lung cancer, we found that the expression of PRIM2 was significantly up-regulated in lung cancers with altered p53 and RB signaling pathways, suggesting that PRIM2 may be regulated by p53 and RB pathways’ It looks like a description of the results, relatively part of the discussion, and there is this paragraph in the introduction. Is this the way it should be?

Response: We appreciate the reviewer for pointing out this important issue. We are sorry for that we did not make it clear in present manuscript. As suggested by the reviewers, we have revised it in the revised manuscript.

6) Line 109-111: if it is liteterure report, citation is missing.

Response: We appreciate the reviewer for pointing out this important issue. We are sorry for that we did not make it clear in present manuscript. This result is not derived from published articles, but TCGA cBioportal platform (https://www.cbioportal.org/results/oncoprint?cancer_study_list=luad_tcga&Z_SCORE_THRESHOLD=2.0&RPPA_SCORE_THRESHOLD=2.0&profileFilter=mutations%2Cgistic&case_set_id=luad_tcga_cnaseq&gene_list=TP53%252C%2520PRIM2&geneset_list=%20&tab_index=tab_visualize&Action=Submit).

7) Line 113-119: again, wording and summaries specific to the results or discussion.

Response: We appreciate the reviewer for pointing out this important issue. We are sorry for that we did not make it clear in present manuscript. As suggested by the reviewers, we have revised it in the revised manuscript.

8) Line 152: ‘2.6. Immunohistochemistry’: How many patients were tested? Where did the study material come from? Was the study prospective or retrospective? The expression of which proteins was determined by this method? Was there approval for the study from the appropriate bioethics committee?

Response: We appreciate the reviewer for pointing out this important issue. We are sorry for that we did not make it clear in present manuscript. We have supplemented patient-related information in the Materials and Methods section. We detected the expression level of PRIM2 protein in 27 lung cancer patients by immunohistochemistry. This study is prospective work, and the study protocol was approved by the Institute Research Ethics Committee at Nankai University.

9) Line 167-176: This paragraph is an introduction and not a description of the results.

Response: We appreciate the reviewer for pointing out this important issue. We are sorry for that we did not make it clear in present manuscript. As suggested by the reviewers, we have revised it in the revised manuscript.

10) Line 183-184: ‘…which may suggest that the abnormal expression of PRIM2 protein is caused by the increased transcription level.’ Again, a phrase that fits the discussion not the results. Further part of the results also contain wording charateristic to the discussion, not the results.

Response: We appreciate the reviewer for pointing out this important issue. We are sorry for that we did not make it clear in present manuscript. As suggested by the reviewers, we have revised it in the revised manuscript.

11) Line 190: KIRC – no expansions of the abbreviation, the same line 218 PCNA. Throughout the article: if authors use abbreviations, the full name of the abbreviation should be given the first time it is used.

Response: We appreciate the reviewer for pointing out this important issue. We are sorry for that we did not make it clear in present manuscript. As suggested by the reviewers, we have revised the full text in the revised manuscript.

12) Line223-225: ‘PCNA is a cofactor for the DNA polymerase delta localized in the nucleus, increases the processivity of the leading strand during DNA replication, and is also involved in DNA repair. Research has found that PCNA is regulated by miRNAs to promote the growth and proliferation of lung cancer [23, 24].’ Once again, sentences specific to the introduction, not the results. The same line 260-262.

Response: We appreciate the reviewer for pointing out this important issue. We are sorry for that we did not make it clear in present manuscript. As suggested by the reviewers, we have revised it in the revised manuscript.

13) The authors identified p53 mutant and non-mutant groups in their analyses. What were the selection criteria? What variants were considered? Were these mutations of pathogenic status, or was selection based on the functional type of mutation? There is no such explanation in the text.

Response: We appreciate the reviewer for pointing out this important issue. We are sorry for that we did not make it clear in present manuscript. After our further analysis, we were unable to define the mutation type of p53, so we removed this result.

14) Line 276-282: The entire paragraph is described as in the discussion. Only here does the reader learn that TP53 expression was determined using IHC.

Response: We appreciate the reviewer for pointing out this important issue. We are sorry for that we did not make it clear in present manuscript. After further thinking, we decided that the results of IHC could not represent the mutation status, and we decided to remove the results.

15) And returning to point 13 of this review, it is only in the course of reading and comparing the graphs that the reader can infer for himself how the selection of patients was done in terms of TP53 mutations. The authors completely overlooked this in the methodology.

Response: We appreciate the reviewer for pointing out this important issue. We are sorry for that we did not make it clear in present manuscript. We have revised it in the revised manuscript.

16) Line 309: ‘Cell growth and proliferation are required for normal cells and tumor cells.’ It is not entirely clear what the authors intended to bring to the discussion with this sentence. Perhaps the authors would consider starting the discussion differently? How about eliminating The following sentences are more precise.

Response: We appreciate the reviewer for pointing out this important issue. We are sorry for that we did not make it clear in present manuscript. We have revised it in the revised manuscript.

17) The authors frequently mention the role of TP53 and RB1 pathways in the article, but did not perform any analysis of RB1. This is somewhat lacking. At the same time, it is understandable that analyses have their limitations in quantity. Nevertheless, RB1 was dragged into the topic of the article, and it would have been nice if there had been an analysis at least at the RB1 mRNA level, if it is often mentioned in the article.

Response: We appreciate the reviewer for pointing out this important issue. We are sorry for that we did not make it clear in present manuscript. In this study, we mainly focused on the regulation of PRIM2 by p53, because we found that the mutation frequency of p53 in lung cancer was much higher than that of RB1. Nevertheless, we also added the mutation frequency of RB1 in lung cancer.

18) The figures are of very good quality, are legible and easily interpretable. Nice.

Response: We appreciate the reviewer for all comments and suggestions.

Round 2

Reviewer 1 Report

The authors have carefully addressed my corners. I recommend acceptance in the present form.

This manuscript is a resubmission of an earlier submission. The following is a list of the peer review reports and author responses from that submission.